# CT radiomics for differentiating fat poor angiomyolipoma from clear cell renal cell carcinoma: Systematic review and meta-analysis

**Fatemeh Dehghani Firouzabadi[1], Nikhil Gopal[2], Amir Hasani[1], Fatemeh Homayounieh[1], Xiaobai Li[3], Elizabeth C. Jones[1], Pouria Yazdian Anari[1], Evrim Turkbey[1], Ashkan A. Malayeri[1] ***

1 Radiology Department, National Institutes of Health, Clinical Center (CC), Bethesda, Maryland, United States of America, 2 Urology Department, National Cancer Institutes (NCI), Clinical Center, National Institutes of Health, Bethesda, Maryland, United States of America, 3 Biostatistics and Clinical Epidemiology Service, NIH Clinical Center, Bethesda, MD, United States of America

* ashkan.malayeri@nih.gov

**Data Availability Statement:** All relevant data are within the paper and its Supporting Information files.

## Abstract

### Purpose

Differentiation of fat-poor angiomyolipoma (fp-AMLs) from renal cell carcinoma (RCC) is often not possible from just visual interpretation of conventional cross-sectional imaging, typically requiring biopsy or surgery for diagnostic confirmation. However, radiomics has the potential to characterize renal masses without the need for invasive procedures. Here, we conducted a systematic review on the accuracy of CT radiomics in distinguishing fp-AMLs from RCCs.

### Methods

We conducted a search using PubMed/MEDLINE, Google Scholar, Cochrane Library, Embase, and Web of Science for studies published from January 2011–2022 that utilized CT radiomics to discriminate between fp-AMLs and RCCs. A random-effects model was applied for the meta-analysis according to the heterogeneity level. Furthermore, subgroup analyses (group 1: RCCs vs. fp-AML, and group 2: ccRCC vs. fp-AML), and quality assessment were also conducted to explore the possible effect of interstudy differences. To evaluate CT radiomics performance, the pooled sensitivity, specificity, and diagnostic odds ratio (DOR) were assessed. This study is registered with PROSPERO (CRD42022311034).

### Results

Our literature search identified 10 studies with 1456 lesions in 1437 patients. Pooled sensitivity was 0.779 [95% CI: 0.562–0.907] and 0.817 [95% CI: 0.663–0.910] for groups 1 and 2, respectively. Pooled specificity was 0.933 [95% CI: 0.814–0.978] and 0.926 [95% CI: 0.854–0.964] for groups 1 and 2, respectively. Also, our findings showed higher sensitivity and specificity of 0.858 [95% CI: 0.742–0.927] and 0.886 [95% CI: 0.819–0.930] for detecting

**Funding:** This research was supported in part by the intramural Research Program of the NIH, Clinical Center.

**Competing interests:** The authors have declared that no competing interests exist.

ccRCC from fp-AML in the unenhanced phase of CT scan as compared to the corticomedullary and nephrogenic phases of CT scan.

## Conclusion

This study suggested that radiomic features derived from CT has high sensitivity and specificity in differentiating RCCs vs. fp-AML, particularly in detecting ccRCCs vs. fp-AML. Also, an unenhanced CT scan showed the highest specificity and sensitivity as compared to contrast CT scan phases. Differentiating between fp-AML and RCC often is not possible without biopsy or surgery; radiomics has the potential to obviate these invasive procedures due to its high diagnostic accuracy.

## Introduction

According to GLOBOCAN data from 2018, an estimated 403,000 persons are diagnosed with kidney cancer each year, accounting for 2.2 percent of all cancer cases [1]. In recent decades, the increased use of abdominal imaging has led to a rise in the number of small renal incidentalomas [2]. Although the majority of these lesions are renal cell carcinomas (RCCs), 20% of them may be benign, with the most common subtype being angiomyolipoma (AML) [3,4]. Renal angiomyolipoma (AML) consists of blood vessels, smooth muscle, and adipose tissue [5]. AMLs can thus generally be diagnosed radiographically on the basis of fat in their lesions. Nevertheless, about 5% of renal AMLs have scant amount of fat, creating a diagnostic dilemma in differentiating between these so-called fat-poor angiomyolipomas (fp-AMLs) and RCC, the latter of which is most commonly clear cell (75% of all RCC cases) [6,7]. Many imaging properties of fp-AML and ccRCC are similar, notably in homogenous ccRCC (hm-ccRCC, regarded as ccRCC without apparent necrosis, cyst, or hemorrhage). Often, tissue procurement, either through biopsy or surgery, is needed for confirmation of suspected diagnosis in these situations [5].

Quantitative texture analysis is a branch of radiomics that extracts and evaluates features from digital images, detects subtle changes and heterogeneity that human vision misses, and provides an objective method in analyzing the intensity, distribution, and relationship of pixels within a digital image [8]. Due to the volume and complexity of generated data, machine learning is often utilized to develop a prediction model based on these image features. Recently, CT radiomics has been applied to differentiate fp-AML from RCC. We conducted a systematic review and meta-analysis of such studies to characterize the performance of CT radiomics in regards to this diagnostic conundrum and thereby determine whether this analysis should be integrated into the clinical setting.

## Methods

### Search strategy

We searched PubMed/MEDLINE, Google Scholar, Cochrane Library, Embase, and Web of Science for papers published in English between January 2011–2022. In a search approach that incorporated a combination of keywords and medical subject headings (MeSH)/EMTREE terms, the following phrases were used: (Radiomics OR Artificial Intelligence OR Machine Learning OR Deep Learning) AND (Kidney OR Renal) AND (angiomyolipoma OR AML).

## Eligibility criteria

Studies were chosen based on the following criteria: (1) Full-text original papers on CT radiomics clinical investigations, including diagnostic case-control trials, to examine the accuracy of differentiating fp-AML and RCCs, (2) Absolute numbers of patients with true positive (TP), false positive (FP), true negative (TN), and false negative (FN) results that were either present or could be calculated. (3) Patients' histopathological data was available, which served as a gold standard for comparing model performance.

## Study selection and data extraction

The papers were assessed independently by the two authors (NG, FH) using Covidence systematic review software (Veritas Health Innovation, Melbourne, Australia, available at www.covidence.org) based on the titles and abstracts; those that did not match the inclusion criteria were discarded. Any discrepancies between the two screeners in included articles were resolved by discussion and submission to a third author (FDF). Following this initial step, two authors (NG, FDF) independently assessed the full texts of all remaining papers for inclusion or exclusion in the final study. References in review articles were checked, and the fulltext was assessed for those that had not previously been screened but were deemed to be pertinent. Conflicts were addressed in the same manner as they were during the initial screening. Each research's study characteristics were retrieved by two writers (FH, FDF). Author, year of publication, AI model, machine learning method, study design, accuracy, validity, sensitivity (SEN), specificity (SPE), PPV, NPV, and area under the curve (AUC) were all included in the data.

## Quality assessment of included studies

We used modified questions from the Quality Assessment of Diagnostic Accuracy Studies 2 (QUADAS-2) criteria because no existing quality assessment tool focused on studies utilizing machine learning [9].

## Data analysis

For identifying fp-AML and RCCs, published data was extracted and translated into a datasheet utilizing the reported or calculated SEN and SPE. The pooled SEN and SPE with 95 percent confidence intervals for model diagnosis were obtained using a random effects model for the meta-analysis. Forest plots and risk of bias graphs were created using datasets. A funnel plot and symmetry analysis as well as Deek's test were used to assess the risk of publication bias. R software, version 3.5.1, was used to conduct all statistical analyses (R Foundation for Statistical Computing, Vienna, Austria, 2018).

## Assessment of heterogeneity

Statistical heterogeneity was first assessed informally using the forest plots of study estimates, and then more formally using the I2 statistic (I2> 50% indicates considerable heterogeneity). The hierarchical summary receiver operating characteristic curve was used to depict all of the studies as a circle. The included studies' heterogeneity was examined. In general, an I2 test with a I2 >50%, p<0.05 indicated considerable heterogeneity.

## Results

### Research and selection of studies

A total of 250 articles were initially identified from all databases using the search terms as described above. 105 duplicate articles were excluded. Additionally, 60 articles were removed after reading their titles and abstracts and being deemed irrelevant. Subsequently, after reading the full texts, 69 articles were found to be reviews or to have no relevant data, and six articles were unavailable for data extraction. Ultimately, ten articles were included. Fig 1 summarizes the article selection process.

### Quality assessment and publication bias

The included studies' quality was assessed using the QUADAS-2 checklist, and the findings are given in Fig 2. Overall, the quality of included studies was satisfactory. In terms of 'risk of

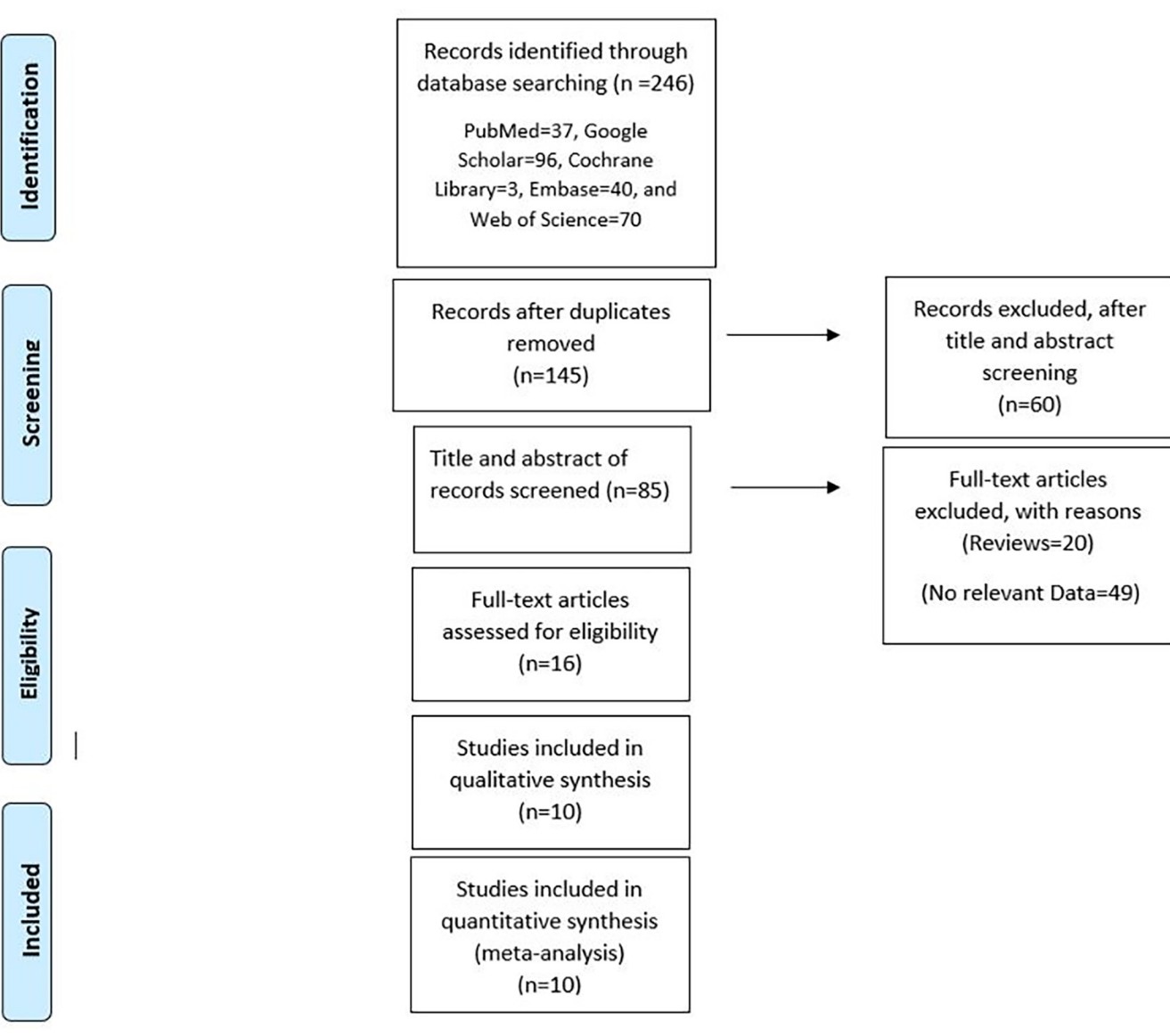

**Fig 1. Prisma chart for included studies.**

**A**

| Authors | D1Q1 | D1Q2 | D1Q3 | D1A | D2A1 | D2A2 | D2A | D3Q1 | D3Q2 | D3A | D4Q1 | D4Q2 | D4Q3 |
|---|---|---|---|---|---|---|---|---|---|---|---|---|---|
| Takahashi | Low | Unclear | Low | Low | Low | Unclear | Low | Low | Low | Low | Unclear | Low | Low |
| Hodgdon | Low | High | Low | Low | Low | Unclear | Low | Low | Low | Low | Low | Low | Low |
| Feng | Low | Unclear | Low | Low | Low | Low | Low | Low | Low | Low | Low | Low | Low |
| Deng | Low | Unclear | Low | Low | Low | Low | Low | Low | Low | Low | Low | Low | Low |
| Yang | Low | High | Low | Low | Low | Unclear | Low | Low | Low | Low | Unclear | Low | Low |

**B**

| Authors | D1Q1 | D1Q2 | D1Q3 | D1A | D2A1 | D2A2 | D2A | D3Q1 | D3Q2 | D3A | D4Q1 | D4Q2 | D4Q3 |
|---|---|---|---|---|---|---|---|---|---|---|---|---|---|
| Nie | Unclear | High | Low | Low | Low | Unclear | Low | Low | Low | Low | Unclear | Low | Low |
| You | Low | Unclear | High | Low | Low | Low | Low | Low | Low | Low | Low | Unclear | Low |
| Ma | Unclear | Unclear | High | Low | Low | Low | Low | Low | Low | Low | Low | Low | Low |
| Cui | Low | High | Low | Low | Low | Unclear | Low | Low | Low | Low | Unclear | Low | Low |
| Yi Yang | Low | Unclear | Low | Low | Low | Low | Low | Low | Low | Low | Low | Low | Low |

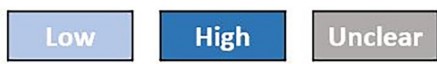

Low  High  Unclear

**Fig 2. Summary of QUADAS-2 assessments of included studies.**

bias' concerns, it was discovered that 'patient selection' exhibited significant flaws, which could indicate bias in terms of inclusion (2 studies in group 1 and 4 studies in group 2). Based on Deek test, there were no publication bias between studies.

### Heterogeneity test

Heterogeneity was tested using Cochran-Q and I2 (S1–S4 Figs). These results indicated high heterogeneity in pooled sensitivity and specificity among the included studies when we used the best performance of model in differentiating fp-AML from RCCs and fp-AML from ccRCC, but there is low heterogeneity between ccRCC from fp-AML when classified based on unenhanced and enhanced CT scan phases in three categories of unenhanced CT scan, corticomedullary, and nephrographic phases of CT scan (S1 and S2 Figs).

### Study characteristics

The characteristics of the included studies are shown in Table 1. All ten studies were retrospective cohort studies. The total number of patients was 1437 with 1456 lesions observed. The average age of patients in studies ranged between 47 and 63 years. In the 7 studies that had information about CT slice thickness, 71% (n = 5) had slice thickness of 1–3 mm, with the remainder utilizing CT with 5 mm slice thickness. The results of the meta-analysis are presented in S1–S4 Figs. Pooled sensitivity, specificity, and odds ratio for detection of fp-AML from RCCs were 0.779 [95% CI (confidence interval): 0.562–0.907], 0.933 [95% CI: 0.814–0.978], and 41.822 [95% CI: 12.788–136.777], respectively (S1 and S2 Figs). Pooled sensitivity, specificity, and odds ratio for detection of fp-AML from ccRCCs specifically were 0.817 [95% CI: 0.663–0.910], 0.926 [95% CI: 0.854–0.964], and 58.6729 [95% CI: 15.125–227.607],

**Table 1. Characteristics of 10 related studies in the meta-analysis.**

| Author | Year | All MLA Algorithms in studies | Methods | Number of Patients | Number of Lesions | Segmentation Method | Segmented region (2D (Largest diameter), 3D (whole volume)) | Single institution/ multicenter | Feature selection– Number (first, higher order, shape, etc.) | CT Phase | MLA-classifier | Sensitivity (sd if mean available) | Specificity (sd if mean available) |
|---|---|---|---|---|---|---|---|---|---|---|---|---|---|
| **Fp-AML vs RCCs** | | | | | | | | | | | | | |
| Takahashi | 2014 | SVM | Radiomics | Total: 153 fp-AML:23 | Total: 172 fp-AML: 24 ccRCC: 98, pRCC:36, other:14 | Manually | 2D | Single | Mean gray-level, angular second moment, gray-level entropy, sum entropy, and sum average | UP | SVM | 83 | 90 |
| Hodgdon | 2015 | LR | Radiomics | Total: 84 fp-AML:16, ccRCC: 51, pRCC:13, chRCC: 20 | | Manually | 2D | Single | gray-level histogram mean and variance, GLCM, GLRLM, Entropy | UP | LR | 88 | 75 |
| Feng | 2018 | SVM | Radiomics | Total: 58 fp-AML: 17 RCC: 41 (31 cc, 6 Chromophobe, two papillary, two multilocular cystic RCC) | Total: 58 fp-AML: 17 RCC: 41 | Manually | 2D | Single | nine histogram features, five Haralick texture features (energy, contrast, correlation, homogeneity and entropy). | Three-phase CT scan | SVM-RFE +SMOTE | 87.8 | 100 |
| Y. Deng | 2019 | LR | Radiomics | Total: 354 fp-AML: 31, oncocytoma: 111 ccRCC: 244, pRCC:46, chRCC:56 | Total: 354 fp-AML: 31, oncocytoma: 116, ccRCC: 249, pRCC:49, chRCC:56 | Manually | 2D | Single | Entropy, maximum positive pixel | Portal venous phase | | 83 | 82 |
| Ruimeng Yang | 2020 | LR, SVM, Naïve Bayes, KNN, Decision tree, Bagging, RF, AdaBoosting | Radiomics | Total: 163 fp-AML: 45 RCC: 118 (Chromophobe: 13, CC: 95, PC: 10) | Total: 163 fp-AML: 45 RCC: 118 (Chromophobe: 13, CC: 95, PC: 10) | Manually | 2D | Single | 1 shape feature, 5 first-order statistics features, and 4 texture features | UP | SVM | 83 | 78 |
| **Fp-AML vs ccRCC** | | | | | | | | | | | | | |
| Cui | 2019 | SVM | Radiomics | Total: 171 fp-AML: 41 ccRCC: 80, pRCC:22, chRCC: 26 | Total: 171 fp-AML: 40 ccRCC: 82, pRCC:22, chRCC: 26 | Manually | 2D | Single | first-order features; shape features; GLCM, GLSZM, GLRLM, NGTDM, GLDM | Three-phase CT scan | SVM-RFECV, SMOTE | 95 | 91 |
| Pei Nie | 2019 | LR | Radiomics | Total: 99 fp-AML: 36 ccRCC: 63 | Total: 99 fp-AML: 36 ccRCC: 63 | Manually | 2D | Single | 14 features related to intensity, shape, and homogeneity were selected as the significant features to build the radiomics signature. Most of the selected features (12/14) were high-order filter and wavelet features | CMP and NP | LR | 96 | 76.67 |
| M.-W. You | 2019 | SVM | Radiomics | Total: 67 fp-AML: 17 ccRCC: 50 | Total: 67 fp-AML: 17 ccRCC: 50 | Manually | 2D | Single | histogram, GLCM, GLRLM | four-phase CECT can | SFS, SVM | 82 | 76 |

*(Continued)*

**Table 1.** (Continued)

| Author | Year | All MLA Algorithms in studies | Methods | Number of Patients | Number of Lesions | Segmentation Method | Segmented region (2D (Largest diameter), 3D (whole volume) | Best performance of model (best performing combination of model) | | | | | | |
| | | | | | | | | Single institution/ multicenter | Feature selection– Number (first, higher order, shape, etc.) | CT Phase | MLA-classifier | Sensitivity (sd if mean available) | Specificity (sd if mean available) |
| **Yi Yang** | 2019 | RF, kNN, SVM, AdaBoost, and sRBFNN | Radiomics | Total: 60 fp-AML: 18 ccRCC: 42 | Total: 60 fp-AML: 18 ccRCC: 42 | Manually | 3D | Single | 18 first-order statistical features, 68 texture features, and 688 wavelet features. | Delayed phase | sRBFNN | 66.67 | 100 |
| **Yanqing Ma** | 2021 | LR | Radiomics | Total: 230 fp-AML: 58 ccRCC: 172 | Total: 230 fp-AML: 58 ccRCC: 172 | Manually | 2D | Single | GLCM entropy– angle-offset, sum variance. High gray level run emphasis- all direction- offset1-SD, run length Non- uniformity-All Direction- offset7-SD, sphericity | CMP, NP | LR | 65.52 | 84.88 |

MLA, machine learning algorithm; CNN, Convolutional Neural Networks; FN, false negative; FP, false positive; TN, true negative; TP, true positive; SVM, support vector machine; UP, Unenhanced phase; CMP, corticomedullary phase; NP, nephrographic phase; GLCM, gray-level co-occurrence matrix; GLSZM, gray-level-size zone matrix; GLRLM, gray-level run-length matrix; NGTDM, gray-tone difference matrix; GLDM, gray-level-dependence matrix; support vector machine with the recursive feature elimination method based on fivefold cross-validation (SVM-RFECV); synthetic minority oversampling technique(SMOTE), SVM-RFE+SMOTE, SVM support vector machine classifier based on candidate feature set, SVM-RFE support vector machine classifier based on optimal feature subset selected by recursive feature elimination, SMOTE synthetic minority over-sampling technique, Sequential feature selection (SFS), sparse radial basis function neural network (sRBFNN).

respectively (S3 and S4 Figs). Also, our findings showed higher sensitivity and specificity of 0.858 [95% CI: 0.742–0.927] and 0.886 [95% CI: 0.819–0.930] for detecting ccRCC from fp-AML in the unenhanced phase of CT scan as compared to the corticomedullary (CMP) and nephrographic (NP) phase of CT scans, respectively (0.755 [95% CI: 0.628; 0.850] and 0.781 [95% CI: 0.656; 0.870] for sensitivity, and 0.882 [95% CI: 0.814; 0.927] and 0.832 [95% CI: 0.734; 0.899] for specificity, respectively).

## Discussion

Renal cell carcinoma (RCC) is the ninth most common cancer worldwide, and is often diagnosed as an incidental mass detected on cross-sectional imaging. In this setting, RCC needs to be distinguished from benign lesions such as AML, which can be difficult based on imaging alone, as in the case of fp-AML. In order to avoid the morbidities of biopsy or surgery, improvements in our ability to non-invasively diagnose fp-AML is warranted, with quantitative texture analysis emerging as a promising method [5].

To our knowledge, this study represents the first meta-analysis of CT radiomics for differentiating between fp-AML and RCCs, with additional subgroup comparison between fp-AML and ccRCC. The results indicate that CT radiomics has a high diagnostic performance in differentiating between fp-AML and RCCs (sensitivity and specificity of 0.779 and 0.933, respectively) as well as between fp-AML and ccRCC (sensitivity of 0.817 and specificity of 0.926, respectively). When we used the best model performance in distinguishing ccRCC from fp AML, we found considerable heterogeneity in pooled sensitivity and specificity among the included studies, but lower heterogeneity when we classified based on unenhanced and enhanced CT phases. Also, we found that there was no significant publication bias between studies.

Previous studies have noted that the majority of the best feature subsets were from the unenhanced (UP) and NP CT phases. As Yang et al [10] showed, the highest AUC; accuracy; sensitivity; and specificity (0.90, 0.82, 0.83, and 0.82, respectively) for fp-AML and RCCs discrimination was the combination of both UP and NP CT phases with the discriminative model of combination of SVM with Relief. They used eight robust classifiers,and 28 feature selection approaches were merged, providing 224 alternative combinational discrimination models for evaluations. Additionally, texture analysis was used by Hodgdon et al. [11] to determine the accuracy of texture analysis in distinguishing fp-AML from RCC using unenhanced CT, and they found an AUC of 0.89. With UP employing texture analysis for discriminating of fp-AML and ccRCC, Yan et al. [12] found a tendency toward better lesion categorization (misclassification rates of 0.00–5.81 percent). Our finding showed better pooled sensitivity and specificity once we classified based on UP and enhanced CT phases (CMP, NP) in the subgroup analysis (0.86, 0.75, and 0.78 for sensitivity, 0.89, 0.88, and 0.83 for specificity, respectively).

A variety of algorithms were used in studies to analyze texture analysis data, including support vector machine (SVM), logistic regression models, random forest (RF), k-nearest neighbor (kNN), sparse radial basis function neural networks (sRBFNN), sequential feature selection, and different logistic classifiers [1–14]. The greatest overall performances of five classifiers including RF, kNN, SVM, AdaBoost and sRBFNN were reported in a study by Yang et al. For delayed phase CT scan images, sRBFNN has the greatest AUC (0.92) and specificity (100), whereas AdaBoost has the highest accuracy (91.67) and sensitivity (83.33) for arterial phase CT images. A specificity of 100 percent is significant since it suggests that all cases of ccRCC are accurately identified, implying that images with ccRCC will receive appropriate treatment. Lee et al. [13] showed that in distinguishing fp-AML from clear cell ccRCC, kNN

and SVM classifiers using ReliefF feature selection approach beat the other combinations, with an AUC of 0.782 and 0.717, respectively.

Takahashi et al. [14] also found that fp-AML has lower entropy than RCC and developed a logistic regression model to distinguish small fp-AML from RCC with sensitivity and specificity of 50% and 98 percent, respectively. High sensitivity and specificity were reported in the study by Feng et al that showed the machine learning classifier from three-phase images had a high sensitivity and specificity of 87.8 and 100%, respectively based on CT texture features including nine histogram features and five Haralick texture features (energy, contrast, correlation, homogeneity and entropy). SVM classifiers enhanced the sensitivity of categorization in Feng et al. investigation when compared to Takahashi et al results. Despite the fact that the SVM-RFE classifier had a reasonably good accuracy, the classifier's low sensitivity meant that patients with minor fp-AML could be misdiagnosed or possibly endure unneeded surgery. The imbalance of the dataset in Feng et al. study was thought to be the primary culprit for the low sensitivity. In an unbalanced dataset, the SVM algorithm tends to sacrifice the minority group in order to gain higher accuracy, resulting in fp-AML instances being misclassified. In previous studies by Feng et al. [15] and Cui et al. [16], synthetic minority oversampling technique (SMOTE) were used to resample the fp-AML group so that their sample size were equivalent to that of the RCC and ccRCC groups, resulting in a rebalanced dataset and reported a high sensitivity and specificity (87.8 and 95% for sensitivity, and 100 and 91% for specificity, respectively) [17]. The combination of SMOTE with an SVM-RFE classifier increased sensitivity while maintaining high specificity, resulting in fewer misdiagnoses and improved classification performance. Nevertheless, the SMOTE process only generates new samples from the existing dataset, which could result in overfitting of data and thus poor external validation.

Improved sensitivity and specificity in detecting fp-AML from ccRCC compared to fp-AML from RCCs is likely due to differential model performance in discrimination of ccRCCs as compared to non-ccRCCs. For instance, in the study by Cui et al., the machine learning classifier had higher accuracy in discriminating ccRCCs from fp-AMLs (AUC 0.97 and ACC 93.29%) as opposed to non-ccRCCs (AUC 0.89 and ACC 84.38%). Non-ccRCCs are homogeneous and hyperattenuating, much like fp-AML, accounting for model difficulties with classification between these two entities.

Radiomics features refer to different parameters, including histogram, texture, form factor, gray-level co-occurrence matrix (GLCM), and gray-level run-length matrix (RLM) parameters. These features have been used to predict the Fuhrman grade of RCC by using CT-based radiomics features [18]. Previous studies have shown that radiomics analysis is highly accurate in distinguishing between fp-AML from RCCs (Table 1). However, ccRCC tends to have a more inhomogeneous texture [19] and a more rounded appearance shape [20] than fp-AML. Despite this, the histogram analysis of attenuation measurement cannot distinguish fp-AML from RCC with 100% specificity, which means that radiomics-only analysis is not enough for diagnosis making. Nonetheless, the radiomics signature associated with conventional CT analysis has become a major trend.

Although radiomics is a promising and powerful addition to the diagnostic armamentarium, there are limitations in the design and execution of the studies that limit its translation into clinical practice. Primarily, all studies in this meta-analysis were done in a single institution center, with the developed model not tested on an outside dataset. The lack of external validation thus limits generalizability of models; indeed, their performances may be overstated due to overfitting. Low sample size, generally seen in these studies, particularly with the proportion of fp-AMLs, increases the likelihood of overfitting. The studies in this meta-analysis utilized different imaging modalities and machine learning algorithms; we recognize that combining heterogenous study designs for a pooled analysis may create undue noise, affecting its

validity. At present however, components of the workflow for radiomic analysis (image acquisition, segmentation, feature selection, machine learning algorithm selection and implementation, and statistical analysis) are not standardized. An additional limitation in this meta-analysis was the small number of studies considered, as we had to exclude studies which were comparing different subgroups and reporting different outputs in their results i.e. mean sensitivity and specificity. In addition, we excluded studies which used MRI to detect these categories. Lastly, due to the limited number of studies available in this area and the absence of comprehensive studies on radiomics features, we were unable to perform a meta-analysis of each category and CT radiomics feature. Moreover, it was difficult to compare CT radiomics features, including first-order, higher-order, and shape features, as well as different signs of CT radiomics, such as texture analysis, enhancement pattern, calcifications, size and shape, and attenuation values. This was due to differences in various parameters, such as CT phase, MLA classifier, and sample sizes, across various studies similar to previous meta-analysis in this area [21].

## Conclusion

This study found that CT radiomics has a high degree of accuracy in distinguishing RCCs vs. fp-AML, especially in detecting clear cell RCCs vs. fp-AML Also, an unenhanced CT scan showed the highest specificity and sensitivity as compared to the corticomedullary and nephrogenic phase of CT scan, suggesting that it might be used as a non-invasive tool to offer critical information for detecting fp-AML. However, to translate CT radiomics into clinical practice, standardization of image acquisition; processing; and analysis between institutions is warranted in order to improve sample size and external validation to allow for more robust prediction models.

## Supporting information

**S1 Checklist. PRISMA-P 2015 checklist.**
(PDF)

**S1 Fig. Forest plot for sensitivity of CT radiomics for differentiating between fp-AML from renal cell carcinomas.**
(TIF)

**S2 Fig. Forest plot for specificity of CT radiomics for differentiating between fp-AML from renal cell carcinomas.**
(TIF)

**S3 Fig. Forest plot for sensitivity of CT radiomics for differentiating between fp-AML from clear cell renal cell carcinomas.**
(TIF)

**S4 Fig. Forest plot for specificity of CT radiomics for differentiating between fp-AML from clear cell renal cell carcinomas.**
(TIF)

**S1 Table. Pooled sensitivity, specificity, and odds ratio for differentiating between AMLs without visible fat (fp-AML) from renal cell carcinomas (group 1) and clear cell renal cell carcinomas (group 2).**
(DOCX)

**S2 Table. Pooled sensitivity, specificity, and odds ratio for differentiating between AMLs without visible fat (fp-AML) from clear cell renal cell carcinomas (group 2) based on**

**different CT phases.**
(DOCX)

## Author Contributions

**Conceptualization:** Fatemeh Dehghani Firouzabadi, Nikhil Gopal, Evrim Turkbey, Ashkan A. Malayeri.

**Data curation:** Fatemeh Dehghani Firouzabadi, Fatemeh Homayounieh.

**Formal analysis:** Fatemeh Dehghani Firouzabadi.

**Funding acquisition:** Ashkan A. Malayeri.

**Investigation:** Fatemeh Dehghani Firouzabadi, Ashkan A. Malayeri.

**Methodology:** Fatemeh Dehghani Firouzabadi, Xiaobai Li, Ashkan A. Malayeri.

**Resources:** Fatemeh Dehghani Firouzabadi, Fatemeh Homayounieh.

**Software:** Pouria Yazdian Anari.

**Supervision:** Fatemeh Dehghani Firouzabadi, Nikhil Gopal, Xiaobai Li, Evrim Turkbey, Ashkan A. Malayeri.

**Visualization:** Fatemeh Dehghani Firouzabadi, Elizabeth C. Jones, Ashkan A. Malayeri.

**Writing – original draft:** Fatemeh Dehghani Firouzabadi.

**Writing – review & editing:** Fatemeh Dehghani Firouzabadi, Nikhil Gopal, Amir Hasani, Ashkan A. Malayeri.

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
