## [Decision Letter · Decision Letter 0]

20 Mar 2023

PONE-D-22-18402CT Radiomics for Differentiating Fat Poor Angiomyolipoma from Clear Cell Renal Cell Carcinoma: Systematic Review and Meta-Analysis.PLOS ONE

Dear Dr. Malayeri,

Thank you for submitting your manuscript to PLOS ONE. After careful consideration, we feel that it has merit but does not fully meet PLOS ONE’s publication criteria as it currently stands. Therefore, we invite you to submit a revised version of the manuscript that addresses the points raised during the review process.

Kindly revise your paper keeping in view the suggested guidelines.

We look forward to receiving your revised manuscript.

Kind regards,

Zahid Mehmood, PhD

Academic Editor

PLOS ONE

Journal Requirements:

“This research was supported in part by the intramural Research Program of the NIH, Clinical Center.”

Additional Editor Comments (if provided):

Please revise your paper according to the following suggested changes;

1. The authors need to re-write the abstract again to show the objective of this paper clearly.

2. Introduction section needs to be finished with summary and the contribution should be pointed out.

3. The proposed approach each stage is not discussed in details, nor each stage is interlink appropriately and the discussion is very poor. Please revise it carefully.

4. The author also needs to make a clear conclusion/novelty at the end of each sub-section of methodology.

5. In result section, the authors require to present clear details about the data analysis.

6. The following recent studies on should be discussed in related work;

Data hiding technique in steganography for information security using number theory, 2019

MedDeblur: Medical Image Deblurring with Residual Dense Spatial-Asymmetric Attention, 2022

Stress Estimation Model for the Sustainable Health of Cancer Patients, 2022

Publishing and interlinking covid-19 data using linked open data principles: toward effective healthcare planning and decision-making, 2022

Image authenticity detection using DWT and circular block-based LTrP features, 2019

Reviewers' comments:

Reviewer's Responses to Questions

**Comments to the Author**

1. Is the manuscript technically sound, and do the data support the conclusions?

Reviewer #1: Yes

2. Has the statistical analysis been performed appropriately and rigorously? 

Reviewer #1: Yes

3. Have the authors made all data underlying the findings in their manuscript fully available?

Reviewer #1: No

4. Is the manuscript presented in an intelligible fashion and written in standard English?

Reviewer #1: Yes

5. Review Comments to the Author

Reviewer #1: 1. A common radiologic dilemma is still not avoided for CT imaging to differentiate fat-poor angiomyolipomas (fp-AMLs) from clear cell renal cell carcinomas (ccRCCs) due to the limitation of density of CT images. CT radiomics may improve to present the renal mass characteristics on basis of CT images, and many studies proved that it is a good tool in detecting fp-AMLs from RCCs. This article about the systematic Review and Meta-Analysis of CT Radiomics for Differentiating fp-AMLs from ccRCCs will be benifit to acknowledge the role of CT radiomics.

2. Radiomics is still being a hot topic because it is a promising and powerful addition to the diagnostic armamentarium, and the limitations is evident in the design and execution of the studies and these limit its translation into clinical practice. The distinctive characteristics of CT radiomics are not deep analyzed and not widely acknowleged. This article did not present the different signs of CT radiomics in detail for RCCs and fp-AMLs, and the cited articles are small.

6. PLOS authors have the option to publish the peer review history of their article (what does this mean?). If published, this will include your full peer review and any attached files.

Reviewer #1: No

---

## [Author Response · Author response to Decision Letter 0]

11 Apr 2023

Dear Dr. Zahid Mehmood, 

Thank you so much indeed for your consideration regarding our manuscript entitled “CT Radiomics for Differentiating Fat Poor Angiomyolipoma from Clear Cell Renal Cell Carcinoma: Systematic Review and Meta-Analysis”. Also, we are deeply thankful to the reviewer whose comments improved the quality of our paper. 

Editor Comments:

Please revise your paper according to the following suggested changes;

1. The authors need to re-write the abstract again to show the objective of this paper clearly.

2. Introduction section needs to be finished with summary and the contribution should be pointed out.

3. The proposed approach each stage is not discussed in details, nor each stage is interlink appropriately and the discussion is very poor. Please revise it carefully.

4. The author also needs to make a clear conclusion/novelty at the end of each sub-section of methodology.

5. In result section, the authors require to present clear details about the data analysis.

6. The following recent studies on should be discussed in related work;

Data hiding technique in steganography for information security using number theory, 2019

MedDeblur: Medical Image Deblurring with Residual Dense Spatial-Asymmetric Attention, 2022

Stress Estimation Model for the Sustainable Health of Cancer Patients, 2022

Publishing and interlinking covid-19 data using linked open data principles: toward effective healthcare planning and decision-making, 2022

Image authenticity detection using DWT and circular block-based LTrP features, 2019

Response: Thank you for your comments. Could you please confirm whether these comments are associated with this manuscript since we noticed that the articles you recommended to discuss in the manuscript are not pertinent to our subject matter.

Reviewers' comments:

Reviewer #1: 

1. A common radiologic dilemma is still not avoided for CT imaging to differentiate fat-poor angiomyolipomas (fp-AMLs) from clear cell renal cell carcinomas (ccRCCs) due to the limitation of density of CT images. CT radiomics may improve to present the renal mass characteristics on basis of CT images, and many studies proved that it is a good tool in detecting fp-AMLs from RCCs. This article about the systematic Review and Meta-Analysis of CT Radiomics for Differentiating fp-AMLs from ccRCCs will be benifit to acknowledge the role of CT radiomics.

2. Radiomics is still being a hot topic because it is a promising and powerful addition to the diagnostic armamentarium, and the limitations is evident in the design and execution of the studies and these limit its translation into clinical practice. 

Comment 1. The distinctive characteristics of CT radiomics are not deep analyzed and not widely acknowleged. This article did not present the different signs of CT radiomics in detail for RCCs and fp-AMLs, and the cited articles are small.

Response 1: Thanks for your comment. We acknowledged your concern regarding “analyzing and discussing distinctive characteristics and signs of CT radiomics” by adding one paragraph in discussion and one paragraph in the limitation.

Regarding your comment on the limited number of cited articles, we made a conscious effort to concentrate on studies that met our inclusion criteria from 2011 to 2022, as these were the articles that had undergone the three stages of screening. We were not able to consider any recent studies from June 2022 to March 2023 as they had not been evaluated for inclusion criteria and our manuscript has been under review at PLOS ONE for the past nine months. 

Please let us know if there are any other changes required. 

Thank you so much for your time and consideration.

Best regards,

Ashkan Malayeri

---

## [Decision Letter · Decision Letter 1]

4 Jun 2023

CT Radiomics for Differentiating Fat Poor Angiomyolipoma from Clear Cell Renal Cell Carcinoma: Systematic Review and Meta-Analysis.

PONE-D-22-18402R1

Dear Dr. Malayeri,

We’re pleased to inform you that your manuscript has been judged scientifically suitable for publication and will be formally accepted for publication once it meets all outstanding technical requirements.

Kind regards,

Zahid Mehmood, PhD

Academic Editor

PLOS ONE

Additional Editor Comments (optional):

Accept in current form as authors have addressed all reviewer comments in the revised manuscript.

Reviewers' comments:

Reviewer's Responses to Questions

**Comments to the Author**

1. If the authors have adequately addressed your comments raised in a previous round of review and you feel that this manuscript is now acceptable for publication, you may indicate that here to bypass the “Comments to the Author” section, enter your conflict of interest statement in the “Confidential to Editor” section, and submit your "Accept" recommendation.

Reviewer #1: All comments have been addressed

2. Is the manuscript technically sound, and do the data support the conclusions?

Reviewer #1: Yes

3. Has the statistical analysis been performed appropriately and rigorously? 

Reviewer #1: Yes

4. Have the authors made all data underlying the findings in their manuscript fully available?

Reviewer #1: Yes

5. Is the manuscript presented in an intelligible fashion and written in standard English?

Reviewer #1: Yes

6. Review Comments to the Author

Reviewer #1: (No Response)

7. PLOS authors have the option to publish the peer review history of their article (what does this mean?). If published, this will include your full peer review and any attached files.

Reviewer #1: No

---

## [Editor Report · Acceptance letter]

18 Jul 2023

PONE-D-22-18402R1 

CT Radiomics for Differentiating Fat Poor Angiomyolipoma from Clear Cell Renal Cell Carcinoma: Systematic Review and Meta-Analysis. 

Dear Dr. Malayeri:

I'm pleased to inform you that your manuscript has been deemed suitable for publication in PLOS ONE. Congratulations! Your manuscript is now with our production department. 

Kind regards, 

on behalf of

Dr. Zahid Mehmood 

Academic Editor

PLOS ONE